# CafeQ: Calibration-free Quantization via Learned Transformations and Adaptive Rounding

## Abstract

Post-training quantization is an effective method for reducing the serving cost of large language models, and the standard approach is to use a round-to-nearest quantization level scheme. But this often suffers from large errors due to certain outliers in the weights. Proposed mitigation mechanisms include applying adaptive rounding, random rotation transformations or committing to a post-training target using calibration data. Unfortunately, this reliance on calibration data can be severely limiting in many real-world scenarios as such data may be unavailable or subject to privacy regulations. In this paper, we propose algorithms to optimize transformations and adaptive rounding without access to *any* calibration data. The optimization is achieved by designing a suitable proxy function for the quantization loss without calibration data. To maintain inference efficiency, we perform structured matrix transformations for single matrices. For paired weights that interact directly in the computation graph, we use dual matrix transformations and adaptive rounding methods. We conduct experiments on Gemma 2 models, and observe consistent improvement over the baselines. For Gemma 2 9B quantization, our method improves the average benchmark score from 61.9 to 62.4 for 4-bit quantization and from 52.0 to 60.6 for 3-bit quantization, while adding less than 3% of computation overhead. Furthermore, our method achieves performance comparable to the commonly used GPTQ method, which needs calibration data.

## 1 Introduction

Large language models (Brown et al., 2020; Chowdhery et al., 2022; Thoppilan et al., 2022; Touvron et al., 2023) contain billions or trillions of parameters, which require several gigabytes or even terabytes of memory. For example, storing a single parameter at standard 16-bit precision (e.g., bfloat16) requires 2 bytes. A model with 100 billion parameters would thus require at least 200 GB of memory just to hold its weights, while a 1 trillion parameter model would require 2 TB. This massive size far exceeds the on-chip high bandwidth memory (HBM) available on a single accelerator like a GPU or TPU. Note that to generate even a single token, the model must execute a forward pass, requiring the accelerator to read the entire set of model weights (billions of parameters) from its memory (Davies et al., 2025). The actual mathematical computations for that single token are incredibly fast. However, the processor spends the vast majority of its time idle, waiting for the data to be fetched from HBM, causing memory to be the bottleneck motivating compression of LLMs.

Quantizing weight matrices is a common strategy to reduce the memory loading time, which improves inference efficiency. For example, instead of using 2 bytes for each parameter, quantization can represent the parameter using just 1 byte (8 bits) or even as few as 4 bits. The standard method for quantizing weight matrices in LLMs is *uniform* quantization. In uniform quantization with $N$ bits, the weights are divided into blocks (channels) $\vec{w} = (w_1, \ldots, w_d)$, and then the range of these weights is computed to determine the quantization scale $s = (w_{\max} - w_{\min})/(2^N - 1)$, where $w_{\max} = \max_i w_i$ and $w_{\min} = \min_i w_i$. Then, each

quantized value is computed as

$$\hat{w}_i = s \cdot \left\lceil \frac{w_i - w_{\min}}{s} \right\rfloor + w_{\min}, \tag{1}$$

where $\lceil x \rfloor$ denotes the nearest integer to $x$.

Despite being widely adopted, uniform quantization suffers from the following drawbacks: (1) The error of uniform quantization is determined by the quantization scale, which is heavily affected by the outlier weights in these matrices. (2) Each weight matrix in the transformer network is independently quantized. This does not take any advantage of the computational structure in the transformer block, particularly in the attention modules. To this end, there are several techniques that propose improvements upon uniform quantization. They can be broadly divided into two categories, Quantization-Aware Training (QAT) and Post-Training Quantization (PTQ). In QAT, the model is trained while simulating the effects of quantization (Krishnamoorthi, 2018; Jacob et al., 2018; Esser et al., 2019) and has been successfully applied to enable efficient, integer-only inference in deep learning models. However, QAT requires access to the original training dataset and a costly training pipeline, making it less agile than PTQ, which in contrast, aims to quantize an *already-trained* model with little to no retraining. PTQ techniques can be differentiated along two broad axes. The first axis is the complexity of the quantization scheme, which ranges from the simplest scalar uniform quantization (Frantar et al., 2023) to more complex methods like scalar look-up table (LUT) quantization (Kim et al., 2024), vector uniform quantization (Tseng et al., 2024a), and vector LUT quantization (Tseng et al., 2024c). The second axis pertains to whether calibration data is needed at all.

*Goals of this paper.* We focus on *calibration data-free* scalar uniform quantization. The rationale for choosing uniform quantization is because it is highly efficient and broadly supported by most modern hardware accelerators. Calibration data-free (or calibration-free, for short) quantization is useful in several scenarios. First, in many applications, representative data for calibration is unavailable. This is particularly true when deploying models for tasks in low-resource languages with limited digital footprints, where acquiring a suitable calibration set is often infeasible (Magueresse et al., 2020). Second, even when data exists, its use may be prohibited due to privacy and security concerns. Models operating on sensitive information, such as protected health information in medical applications or biometric data for identification, cannot easily use this data for quantization (Cai et al., 2020). Finally, relying on a static calibration set introduces a significant vulnerability to domain shift. A model may need to be deployed for multiple or unknown downstream tasks, and specifically quantizing for each task may be prohibitive. A model quantized based on a specific dataset can see a degradation in performance when the real-world data it encounters evolves or differs from the calibration set. These motivations underscore the necessity of robust quantization techniques that are *data-independent*, thereby ensuring broader applicability, enhanced privacy, and robustness to domain shifts, in downstream applications.

The central questions in scalar PTQ for LLMs revolve around two main challenges:

**(a) Handling outliers:** How do we prevent a few large magnitude values from dominating the quantization range and destroying precision for all other inliers?

**(b) Rounding:** How to map high-precision values to their low-precision counterparts?

## 2 RELATED WORK

We refer readers to Appendix A for a more comprehensive overview of quantization methods. In this section, we overview related works on calibration-free scalar uniform quantization.

**Handling outliers**: If we use the naive uniform scalar quantization (Equation (1)), which uses the global minimum and maximum of a weight matrix (or sub-matrix), then the error of each parameter would be $O(s)$, which scales linearly in the difference between maximum and minimum values of parameters. However, in large language models this value can be quite large (Chee et al., 2024). To overcome this, Dettmers & Zettlemoyer (2023); Li et al.

(2025a) proposed to use a small number of bits to quantize inliers and a higher number of bits for outliers. A more popular approach is to multiply the matrix with a random matrix prior to quantization (Adepu et al., 2024; Ashkboos et al., 2024b). A random matrix ensures most values are of the same range and reduces the difference between the minimum and the maximum. This is similar to the phenomenon observed in other quantization works such as gradient compression (Suresh et al., 2017; Vargaftik et al., 2022) and JL transform (Vempala, 2005; Kane & Nelson, 2014).

**Rounding techniques:** To the best of our knowledge, all prior calibration-free methods (e.g., Dettmers & Zettlemoyer (2023)) use round to the nearest mapping, where each parameter is mapped to the nearest quantized value. However, we note that there are several results that propose better rounding methods if one assumes that calibration data is available e.g., Frantar et al. (2023); Malinovskii et al. (2024).

Separately, there are other works that propose calibration free quantization by using synthetic or distilled data (Cai et al., 2020; Xu et al., 2020; Sharma et al., 2021). We refer readers to the recent survey of Kim et al. (2025b) for further motivation and techniques in other areas of deep learning. However, these do not focus on LLMs; calibration free quantization methods for LLMs are less explored, and is the focus of our work.

# 3   OUR CONTRIBUTIONS

Before we proceed to our results, we provide a brief review of LLMs. Modern LLMs consist of stacked layers of transformer blocks, and an embedding layer. Each transformer block includes an attention block and a feedforward-network (FF) block, parameterized by the corresponding weight matrices, and scaling vectors in layer norms. The embedding layer is parameterized by an embedding matrix. Most of the computation in these layers is the matrix-vector multiplication $y = Wx$. In this paper, we focus on weight-only quantization that quantizes weight matrices to preserve the result of the output multiplication operations. With this background in mind, we propose a calibration-free PTQ framework that addresses the above mentioned challenges through three primary contributions:

**Handling outliers via proxy-loss minimization** A central challenge in calibration-free quantization is the inability to measure the impact of quantization error on downstream task performance. We first establish that the Frobenius norm of the quantization error, $\|W - \widehat{W}\|_F$ (where $W$ is the original matrix and $\hat{W}$ is the quantized matrix), serves as a good proxy for final task accuracy. Motivated by this correlation, we reframe the problem of mitigating outliers as an optimization task. We focus on methods that use an affine transforms on the weight matrices to mitigate outlier effects. Therefore the question becomes: can we learn an affine transformation that directly minimizes this Frobenius norm error?

**Structured transformations for single matrices** For layers that operate independently, such as feed-forward layers, following earlier works of Adepu et al. (2024); Ashkboos et al. (2024b), we propose to apply an efficient transformation $M$ to the matrix $W$ before quantization and then apply the inverse of the transformation during inference. The effective computation is thus $M^{-1}\widehat{MW}$, where as before $\widehat{\cdot}$ denotes the quantization operation. However, unlike prior works that use fixed or randomized transformations, we learn a structured transformation. This structured transformation is estimated by minimizing the aforementioned proxy loss. We find that block-diagonal matrices perform best in our experiments, in contrast to the Walsh-Hadamard transform used in earlier works.

Structured transformations can be applied to all weight matrices, including the attention block, the feedforward block, and the embedding layer. We will discuss this technique in detail in Section 4.1, and how one can choose $M$ such that the $M^{-1}$ can be applied efficiently.

**Transformations for coupled matrices.** For matrices that are coupled in the computation graph (i.e., they are applied sequentially *without* an intermediate non-linearity), we propose learning an arbitrary matrix $M$. This incurs no additional overhead, as the transformation can be absorbed. If $W_1$ and $W_2$ are coupled matrices, we propose to find $M$ such that the product of the quantized transformed matrices $\widehat{W_1 M^{-1}}\widehat{MW_2}$ closely approximates the original product $W_1 W_2$.

To illustrate the usage of this technique, consider attention score computation between locations $i, j$ below:

$$Attn_{i,j} = \text{Softmax}\left(\frac{x_j^T \cdot W_q \cdot W_k \cdot X_i}{\sqrt{D}}\right) \odot (W_o \cdot W_v \cdot X_i).$$

Note that the attention computation remains unchanged if the product of certain pairs of matrices remain the same, such as the $(W_v, W_o)$ pair and the $(W_q, W_k)$ pair. Note that for any pair of matrices $W_1, W_2$, and any invertible matrix $M$, we have

$$W_1 W_2 = (W_1 M^{-1})(M W_2).$$

Hence no additional online operation is needed since the output of the operation is already preserved. While this technique is incompatible with rotary positional embeddings due to non-commutativity, recent architectures are moving towards using "NoPE", or eliminating positional embeddings altogether (Kazemnejad et al., 2023), given evidence that transformers learn to represent position information from scratch (Haviv et al., 2022). For example, Llama 4 uses attention layers that alternatively use RoPE then NoPE (Meta, 2025). However, since many architectures still use RoPE, we focus on applying our paired quantization technique to the output-value projection pair $(W_v, W_o)$ and leave applications to the query-key pair, $(W_q, W_k)$, for future work.

**Better adaptive rounding techniques for coupled matrices.** For coupled matrices, we also propose a new alternating adaptive rounding technique that accounts for the matrix product structure. Intuitively, if a value in the first matrix is rounded down, our method attempts to compensate by rounding up corresponding values in the second matrix that interact with it during multiplication. Similar to our previous contribution, this can be applied to $W_o$ and $W_v$ matrices in all transformer architectures, and for the query-key pair, $(W_q, W_k)$, in some architectures.

By using a combination of these methods, we observe consistent improvement over other calibration-free baselines on Gemma 2 models. For Gemma 2 9B quantization, our method improves the average benchmark score from 61.9 to 62.4 for 4-bit quantization and from 52.0 to 60.6 for 3-bit quantization on standard benchmarks, while adding less than 3% of computation overhead.

The rest of the paper is organized as follows. In Section 4.1, we propose structured transformations for single matrices, and in Section 4.2, we propose transformations for coupled matrices. In Section 5, we propose our adaptive rounding technique. Finally, in Section 6, we provide ablation studies and results on the Gemma 2 family of models.

## 4 Removing outliers through learned linear transformations

Our quantization approach is motivated by the fact that random rotations remove outliers (Adepu et al., 2024; Tseng et al., 2024a; Ashkboos et al., 2024b), which can reduce quantization error when applied to the weight matrix before uniform quantization. To further remove outliers beyond random rotations, we propose to learn weight-dependent linear transformations.

More precisely, for a weight matrix of shape $d_1 \times d_2$, we learn a weight-dependent invertible transformation matrix $M : d_1 \times d_1$, and perform uniform quantization to $MW$ to get $\widehat{MW}$. During inference, we apply the inverse transformation $M^{-1}$ to the input features $X$ of shape $B \times d_1$ where $B$ is the batch size to get $XM^{-1}$ and the final output of the layer will be

$$\hat{Y} = X M^{-1} \widehat{MW}.$$

Note that compared to the desired output $Y = XW$, the error on the output introduced by the quantization step is

$$\hat{Y} - Y = X M^{-1} \widehat{MW} - XW = X(M^{-1}\widehat{MW} - W).$$

When no quantization is performed, $M^{-1}\widehat{MW} - W = M^{-1}MW - W = \mathbf{0}$, preserving the output. When quantization is performed, we want to learn $M$ such that the expected error

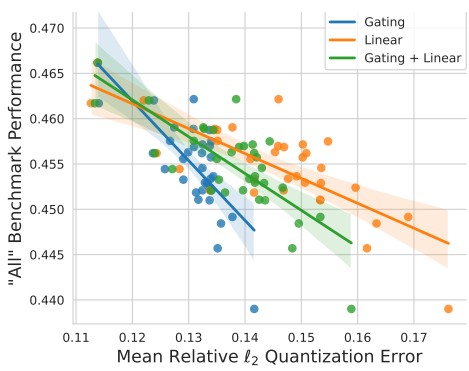

Figure 1: Downstream performance as a function of quantization error of the feed-forward weights. All models were Gemma 2 2B pretrained models whose gating and linear weights were quantized. *Gating + Linear* corresponds to computing the mean quantization error over both sets of weights. We obtain Spearman's rank correlation coefficient of -0.640 (4.6e-5) and -0.679 (10.0e-6) for the gating and linear layers, respectively.

on the outputs $\mathbb{E}_Y[\|Y - \hat{Y}\|_2^2]$ is minimized. With a calibration set, we could approximate the expected error with an empirical average. However, this method cannot be applied in the calibration-free setting. Our approach for learning $M$ relies on the following inequaltiy, which follows from the Cauchy–Schwarz inequality:

$$\mathbb{E}[\|Y - \hat{Y}\|_2^2] = \mathbb{E}[\|X(M^{-1}\widehat{MW} - W)\|_2^2] \le \mathbb{E}[\|X\|_2^2\|M^{-1}\widehat{MW} - W\|_F^2] \tag{2}$$
$$= \|M^{-1}\widehat{MW} - W\|_F^2 \mathbb{E}[\|X\|_2^2].$$

Assuming that the change of the expectation on the input norm $\mathbb{E}[\|X\|_2^2]$ due to quantization is small, we minimize the expected error on the outputs by using $\|M^{-1}\widehat{MW} - W\|_F$ as a proxy. Note that this is the same as the $\ell_2$ loss on the vectorized version of $M^{-1}\widehat{MW} - W$. In this paper, we also refer to this proxy loss as the $\ell_2$ loss.

We perform an empirical study on the feed-forward (FF) block and observe strong Spearman's rank correlation between the proposed $\ell_2$ proxy loss and the scores on downstream evaluations (see Appendix C for discussion on the details of the experiment). The results are presented in Fig. 1. The above finding suggests that the Frobenius norm of the reconstruction error on weight matrices is a good proxy loss to use to calibration-free quantization. A different challenge we face with this transformation-based approach is the extra online computation due to the need for applying $M^{-1}$. In Section 4.1 and Section 4.2, we discuss how this computation can be mitigated for single and coupled matrices.

### 4.1 Learned structured transformation for a single matrix

To reduce the extra inference-time computation, we consider structured matrices that support fast matrix-vector multiplication. More specifically, we use block diagonal matrices. A block diagonal matrix of dimension $d$ and block size $k$ can be written as

$$M = \begin{pmatrix} M_1 & & \\ & \ddots & \\ & & M_{d/k} \end{pmatrix}$$

where all $M_i$'s are $k \times k$ matrices and the neglected entries in $M$ are all zeros. The number of free parameters in a block diagonal matrix is $d \cdot k$. We note that the block diagonal matrices can be multiplied efficiently. During inference, the cost for multiplying such a matrix with a $d$-dimension vector will be $C_{d,k} = O(d \cdot k)$.

**Optimization of $M$.** For optimizing $M$, we initialize $M$ with a block diagonal matrix with each block being a random rotation matrix, and perform gradient descent with $\|M^{-1}\widehat{MW} - W\|_F^2$ as the loss. In Table 1, we present the average relative $\ell_2$ loss we get with different diagonal block sizes on the Gemma 2 2B model and compare it with uniform quantization and transformed quantization with a random structured matrix. As we can see, we are able to get better average $\ell_2$ compared to both UNIFORM and RANDOM* with only small additional FLOPs with diagonal block size 32, and the error gets significantly better as the diagonal block size increases. Furthermore, the block diagonal matrices can be applied

Table 1: The effect of block diagonal size on the down projection layer of Gemma 2 2B (input dimension: 9216, output dimension: 2304). RANDOM denotes applying a block diagonal matrix with block size 1024, and each of the block is a random Hadamard matrix. The additional FLOPs (%) and additional wall-clock time (on V100 GPU) represent the extra compute and time required compared to the original down-projection matmul operation.

| Method | UNIFORM | RANDOM | Learned block diagonal | | | |
|---|---|---|---|---|---|---|
| Diagonal block size | - | 1024 | 32 | 64 | 128 | 256 |
| Avg. relative $\ell_2$ loss | 0.176 | 0.155 | 0.113 | 0.103 | 0.094 | 0.085 |
| Extra FLOPs (%) | 0 | 0.11 | 0.35 | 0.69 | 1.39 | 2.78 |
| Extra wall-clock time (%) | 0 | - | 2.4 | 2.9 | 3.3 | 4.9 |

efficiently on GPUs. In Table 1, we also present the extra FLOPs and wall-clock time needed in an GPU implementation compared to the original computation in the down projection layer. We note that the extra FLOPs is 1.4% and the extra wall clock time is less than 3.3% for a diagonal block size of 128, which is the setting we use in the experiments.

### 4.2 LEARNED PAIRED QUANTIZATION FOR COUPLED MATRICES

As discussed in Section 3, in the attention computation there are pairs of weight matrices that would not affect the attention output as long as their product is preserved. For these cases, we can couple the transformations of the two matrices so that no online operation is needed. Let $W_1 : d_1 \times d_2$ and $W_2 : d_2 \times d_3$ be the weight matrices of two consecutive linear layers, and $M$ be an invertible matrix; then we have for any input to the first layer $X$,

$$XW_1MM^{-1}W_2 = XW_1W_2.$$

Hence, motivated by Eq. (2), we aim to learn $M : d_2 \times d_2$ such that applying quantization on $W_1M$ and $M^{-1}W_2$ results in a small $\ell_2$ error on matrix products, described below.

$$\min_{M, \widehat{\cdot}} \quad \|\widehat{W_1M}\widehat{M^{-1}W_2} - W_1W_2\|_F. \tag{3}$$

We refer to this quantity as the *paired quantization error* (PQE). Another design choice in minimizing PQE is the rounding mechanism, which exploits a joint design. In this section, we focus on learning the transformation matrix $M$ with the aim of removing outliers, and discuss joint design of the rounding mechanism in Section 5.

The goal of the transformation matrix is to mitigate outliers in coupled weight matrices. We formulate the coupled matrix optimization problem as below.

$$\min_{M} \quad \max\{\|MW_1\|_\infty, \|M^{-1}W_2\|_\infty\}$$
$$s.t. \quad M \text{ is invertible.} \tag{4}$$

We want to find an invertible matrix, $M$, that minimizes the maximum absolute value across both transformed weight matrices.

**Pseudo-loss** While Eq. (4) is a reasonable proxy for minimizing intrinsic quantization loss, it is difficult to optimize – only the maximum value entry will be active at each optimization iteration. Thus, we consider pseudo-losses that admit a smoother optimization landscape. We experiment a few choices of pseudo-loss and observe that LogSumExp works the best, described below. For a consideration of the choice of pseudo-loss and other learning hyperparameters, see Appendix D.

Let $U = W_1M$ and $V = M^{-1}W_2$. Let the following be channel-wise maximum absolute values of $U$ and $V$, respectively:

$$m_u(i) = \max_j |U_{i,j}|, \qquad m_v(j) = \max_i |V_{i,j}|$$

We consider the pseudo-loss (1) the LogSumExp over channel-wise maxima:

$$\text{LogSumExp}(U, V) = \log(\sum_{i=1}^{d_1} e^{tm_u(i)} + \sum_{j=1}^{d_3} e^{tm_v(j)}),$$

where $t$ is a temperature hyperparameter.

**Constraints on $M$**   We observe that $M$ need only be invertible to ensure that the transformation is computationally invariant. Prior work learns $M$ such that $M$ is strictly a rotation matrix (Liu et al., 2024). In this work, we varied the degree to which $M$ is orthonormal by minimizing a pseudo-loss directly with Adam without strictly enforcing orthonormality. We added a regularization term to the loss to encourage $M$ to not stray too far from orthonormality. The loss function we minimize is

$$\text{LogSumExp}(MW_1, M^{-1}W_2) + \lambda \frac{\|MM^T - I\|_F}{\sqrt{d}}$$

We note that the loss provides a smooth optimization curve and we obtain lower loss compared to the case where $M$ is strictly constrained to be a rotation matrix. Details on various optimization algorithms and analysis on optimzation dynamics can be found in Appendix D.

## 5 JOINT QUANTIZATION OF WEIGHT MATRICES

In Section 4.2, we described how one can learn a transformation $M$ such that the PQE with individual rounding is minimized. In this section, we modify the rounding map itself ($\widehat{\cdot}$), to additionally minimize the PQE defined in Eq. (3).

The algorithm takes as inputs two weight matrices and a base rounding map Q, which permits fast inference-time dequantization. For example, Q can be uniform quantization or uniform quantization after applying a transformation learned from the previous step. The algorithm starts by quantizing $W_1$ independently. Then it proceeds iteratively, updating each quantized matrix to compensate for the quantization error in the other matrix. More specifically, given a quantized version $\widehat{W}_1$ of $W_1$, instead of quantizing $W_2$, it instead quantizes:

$$W_2' = \widehat{W}_1^\dagger W_1 W_2,$$

where $\widehat{W}_1^\dagger$ is the pseudoinverse of $\widehat{W}_1$, with the goal of having

$$\widehat{W}_1 W_2' = W_1 \widehat{W}_1^\dagger W_1 W_2 \approx W_1 W_2.$$

Then we update the quantized version of $W_2$ as $\widehat{W}_2 = Q(W_2') = Q(\widehat{W}_1^\dagger W_1 W_2)$.

---

**Algorithm 1** Adpative rounding for matrix product.

---

**Input:** Matrices to be quantized $W_1 : d \times h$ and $W_2 : h \times d$, number of iterations $I$, base quantization function Q.
  1: $\widehat{W}_1 \leftarrow Q(W_1)$
  2: **for** $i \leftarrow 1$ **to** $I$ **do**
  3:     $\widehat{W}_2 \leftarrow Q(\widehat{W}_1^\dagger W_1 W_2)$.
  4:     $\widehat{W}_1 \leftarrow Q(W_1 W_2 \widehat{W}_2^\dagger)$.
  5: **end for**
  6: **return** $\widehat{W}_1, \widehat{W}_2$.

---

The above iterative quantization scheme still permits fast inference-time dequantization since both $\widehat{W}_1, \widehat{W}_2$ are obtained by the basic rounding scheme $Q$. Moreover, the iterative procedure can produce even lower error $\ell_2$ quantization error on the matrix product by explicitly compensating for the quantization loss in the other matrix.

Table 2: Average relative $\ell_2$ reconstruction error comparison on $W_v W_o$ with 4 bit per-channel quantization. I stands for independent rounding, and A stands for adaptive rounding. The average is over all Gemma 2 2B layers.

| VO Quantization method | UNIFORM (I) | RANDOM (I) | Learned (I) | Learned (A) |
|---|---|---|---|---|
| Average $\ell_2$ loss | 0.182 | 0.179 | 0.149 | **0.117** |

We provide a toy sample to illustrate this. Suppose one wants to quantize $2 \times 2$ matrices:

$$W = W_1 = W_2 = \begin{bmatrix} 1 & 0 \\ 0 & 0.6 \end{bmatrix}$$

to 1 bit, such that the PQE of $W_1$ and $W_2$ is minimized. Uniform row-wise or column-wise quantization of either matrix will produce: $\widehat{W} = \begin{bmatrix} 1 & 0 \\ 0 & 1 \end{bmatrix} = I$, yielding $\widehat{W_1}\widehat{W_2} - W_1 W_2 = \begin{bmatrix} 0 & 0 \\ 0 & 0.64 \end{bmatrix}$ and a PQE of $\|\widehat{W_1}\widehat{W_2} - W_1 W_2\|_F = 0.64$. Alternatively, applying even a partial iteration of Algorithm 1 yields a reduction in the PQE to 0.36. To see this, in our iterative quantization scheme, we first quantize $\widehat{W_1} \leftarrow Q(W_1) = I$, according to uniform quantization. Updating $\widehat{W_2} \leftarrow Q(\widehat{W_1}^\dagger W_1 W_2) = Q(IW_1 W_2) = Q\left(\begin{bmatrix} 1 & 0 \\ 0 & 0.36 \end{bmatrix}\right) = \begin{bmatrix} 1 & 0 \\ 0 & 0 \end{bmatrix}$, yielding a PQE of $\|\widehat{W_1}\widehat{W_2} - W_1 W_2\|_F = 0.36$.

We conduct experiments to validate the gain of learned matrix and adaptive rounding for a Gemma 2 2B model, and the results are listed in Table 2. The learned transformation improves $\ell_2$ error by 16% over random rotation and independent rounding. The adaptive rounding further reduces it by an additional 21%.

The time complexity of our iterative algorithm at each step is dominated by computing the Moore-Penrose inverse, which itself is dominated by the singular value decomposition (SVD) of each constituent matrix. The time complexity of the SVD of a $d \times h$ matrix is $O(d \cdot h \cdot \min(d, h))$ (Golub & Van Loan, 2013). Therefore, the time complexity of the whole algorithm on input matrices of $d \times h$, number of iterations $I$, and with Q as the uniform quantization operation is $O(I \cdot d \cdot h \cdot \min(d, h))$.

## 6 EXPERIMENTS

We perform experiments with pretrained Gemma 2 2B and 9B models (Team, 2024). We consider both 3 and 4 bit quantization with each quantization block being a channel along the contraction dimension (per-channel quantization) and a subchannel with size 256 and 128. We perform quantization only on the weights and leave the embedding table and activations unquantized. For CafeQ quantization, we use paired transformation for the pair of $W_v$ and $W_o$. For all the remaining matrices, we use block Hadamard matrix along the contraction dimension with diagonal size 128 for Gemma 2 2B and diagonal size 256 for Gemma 2 9B. In total, this adds $< 3\%$ of additional FLOPs for both models.

We evaluate the model on standard Gemma 2 benchmarks (Team, 2024), detailed in Table 6 in the appendix. We present the results in Table 3. As shown, our method achieves superior performance compared to the uniform quantization baselines with no rotation or random Hadamard rotations. The improvement is consistent for both the 4-bit case where the quantization loss is smaller and the 3-bit case where the quantization loss is large. We also observe consistent improvement with different quantization block sizes.

Next we present a few ablation studies we performed on Gemma 2 2B model. All ablations are done with 4 bits and per-channel quantization.

**Ablation on each separate weight matrices.** We study the effect of quantization on each of the components in a transformer block, including FF matrices, QK matrices, and VO matrices. For ablation on each of the component, we keep other components unquantized. For CafeQ, we use a diag block size of 128. The results are presented in Table 4. We see

Table 3: Average benchmark scores for Gemma 2 2B and 9B quantization.

| Method | Gemma 2 2B | | | | | | Gemma 2 9B | | | | | |
| | 3 bits | | | 4 bits | | | 3 bits | | | 4 bits | | |
| | N/A | 256 | 128 | N/A | 256 | 128 | N/A | 256 | 128 | N/A | 256 | 128 |
|---|---|---|---|---|---|---|---|---|---|---|---|---|
| Unquantized | 48.0 | | | | | | 63.9 | | | | | |
| UNIFORM | 23.3 | 33.1 | 34.9 | 41.3 | 44.7 | 45.4 | 28.0 | 50.4 | 58.0 | 58.1 | 61.1 | 62.3 |
| RANDOM | 26.1 | 34.0 | 35.3 | 43.6 | 44.5 | 45.3 | 40.4 | 52.0 | 59.0 | 60.3 | 61.9 | 62.3 |
| CafeQ (ours) | **35.1** | **38.7** | **39.1** | **45.3** | **45.8** | **45.9** | **53.7** | **60.6** | **61.7** | **61.9** | **62.4** | **62.4** |

that our method achieves consistent improvement for each individual component of the transformer block. We see that quantization causes the most score drop on FF blocks, and the least drop on QK blocks. This could be explained by the large parameter count on FF blocks. While QK and VO have similar parameter counts, QK only affects the attention scores, which might have less impact on the overall performance.

| Method \ Weights | FF | QK | VO |
|---|---|---|---|
| Unquantized | | 48.0 | |
| Uniform | 43.9 | 47.2 | 45.9 |
| Random | 45.3 | 47.4 | 46.4 |
| CafeQ | 46.6 | 47.6 | 46.7 |

Table 4: Overall downstream performance after applying uniform/random rotation/CafeQ 4-bit quantization to each set of model weights for pretrained Gemma 2 2B, leaving other weights unquantized. For CafeQ, we use a diag block size of 128.

**The effect of block length.** Next we study how the diagonal block size in the transformation matrix would affect the downstream evaluations. We observe in Table 5 that as the block diagonal size increases, the downstream evaluation gets better overall average score due to the increasing expressiveness of the optimization space. With a block size of 32, we are already improving over the RANDOM transformation-based method.

Table 5: The effect of block diagonal size on average scores on downstream evals. And comparison to calibration-based method GPTQ. We perform per-channel 4-bit quantization on Gemma 2 2B model for all methods.

| Method | UNIFORM | RANDOM* | GPTQ | Learned block diagonal | | | |
|---|---|---|---|---|---|---|---|
| Diagonal block size | - | 1024 | - | 32 | 64 | 128 | 256 |
| Avg. score | 41.3 | 43.6 | 43.7 | 44.7 | 45.4 | 45.3 | 46.2 |

**Comparison to calibration-based method.** In Table 5, we also compare our method with GPTQ (Frantar et al., 2023), a canonical calibration-based method. We use the GPTModel codebase (ModelCloud.ai, 2024) and perform quantization with 1024 calibration samples from the C4 datatset (Raffel et al., 2020) as recommended. We see that the method is able to achieve better average score compared to GPTQ. This shows that learning-based method for removing outliers is important for quantizing LLMs. We view our method and GPTQ as two orthogonal techniques as GPTQ focuses more on how to better round the matrix with calibration data. We view combining the techniques to achieve even better performance as interesting future work.

## 7 CONCLUSION AND DISCUSSION

We propose CafeQ, a calibration-free LLM quantization method for improving uniform quantization with learned transformation matrices and adaptive rounding. We showed that our method consistently improves over other calibration-free baselines while adding minimal extra online computation. We view studying how to combine our method with other calibration-based quantization methods as interesting future directions.

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

## A    RELATED WORK

The idea of transforming weight matrices to make them easier to quantize has been widely explored in the literature. SmoothQuant (Xiao et al., 2023) and OmniQuant (Shao et al., 2024) uses calibrations data to apply activation-dependent scaling on weight matrices.

Applying random rotations to weight matrices has been shown to be an effective way of mitigating outliers prior to quantization. This idea has been explored in recent works to improve the performance of uniform quantization (Adepu et al., 2024; Ashkboos et al., 2024b; Chee et al., 2024; Liu et al., 2024). This has been extended to use vector quantizers instead of scalar quantizers Tseng et al. (2024c;a). Using non-uniform scalar quantizers has been studied in Malinovskii et al. (2024). All these methods use calibration data to design the quantizers. There are methods that do not use rotations, but try to index outliers directly Dettmers et al. (2024); Kim et al. (2024); Li et al. (2025b), but these methods require to index outliers and also need calibration data for good performace. To reduce the inference-time computation to rotate the matrices back, two main techniques have been used: (1) Exploiting the computational invariance property (Ashkboos et al., 2024a;b; Liu et al., 2024; Ma et al., 2024) in transformer blocks to avoid online rotations for "paired", composed, matrices, (2) Rotate the weight matrices using a restricted class of structured matrices (Adepu et al., 2024; Tseng et al., 2024b; Ashkboos et al., 2024b) such as Hadamard matrices, that admit fast online rotations.

Methods have also been proposed to introduce correlation between the quantization for different weights and channels to further reduce quantization error. For example, Frantar et al. (2023); Chee et al. (2024); Kim et al. (2025a) use a calibration dataset to calculate the Hessian matrix of the output error with respect to each model weight, and use the Hessian matrix to introduce correlations between the quantization function of different weights. While most works quantize each layer independently, Liu et al. (2024); Tseng et al. (2024b); Egiazarian et al. (2024) use finetuning to introduce cross-weight dependencies to further reduce the quantization error.

Similar to our work, FlatQuant (Sun et al., 2025) also considers learned structured matrices for better quantization of non-paired matrices. They use Kronecker product of two lightweight matrices for faster online reverse transformation. The concurrent work of FPTQuant (van Breugel et al., 2025) considers learning non-rotation matrices for the transformation matrix on $(W_v, W_o)$ pair. However, all these learning-based methods rely on calibration data for the optimizing the transformation.

## B    EVALUATION DATASETS

Table 6 lists the evaluation tasks considered in this work, along with how they are grouped into task suites.

## C    DETAILS FOR FEEDFORWARD PARAMETER SWEEP

Here we provide sweep details for producing Fig. 1. In this case we considered 4-bit quantization of the gating and linear weights of the feedforward network. We report model performance averaged over *All* downstream tasks, and swept over various quantization hyperparameters in order to produce a range of models with varying intrinsic and extrinsic performance. We considered models without any quantization, vanilla uniform quantization, random rotation/Hadamard transformation prior to quantization, in addition to a range of models with learned block Hadamard transformations. For all learning runs, we set the number of iterations to 20,000, and swept over the following parameters for learning transformations of the FFW:

- Learning rate: $\{0.1, 0.5, 1.0\}$
- Block diagonal size: $\{2, 4, 8, 16, 64, 256\}$
- Number of Hadamard matrices: $\{0, 1, 2\}$

| Task | Core | Math | Code | Notes | Eval Setting |
|------|------|------|------|-------|--------------|
| MMLU (Hendrycks et al., 2021a) | x | x | | 5-shot | scoring |
| ARC (Challenge) (Clark et al., 2018) | x | x | | 0-shot | scoring |
| GSM8K (Cobbe et al., 2021) | x | x | | 8-shot | sampling |
| AGIEval (English) (Zhong et al., 2024) | x | | | 3-5-shot | sampling |
| BBH (Suzgun et al., 2023) | x | x | | 0-shot | sampling |
| Winogrande (Sakaguchi et al., 2021) | x | x | | | scoring |
| HellaSwag (Zellers et al., 2019) | x | | | 0-shot | scoring |
| MATH (Hendrycks et al., 2021b) | | x | | 4-shot | sampling |
| ARC (Easy) (Clark et al., 2018) | | | | 0-shot | scoring |
| PIQA (Bisk et al., 2020) | | | | | scoring |
| SIQA (Sap et al., 2019) | | | | | scoring |
| BoolQ (Clark et al., 2019) | | | | 0-shot | scoring |
| TriviaQA (Joshi et al., 2017) | | | | 5-shot | sampling |
| NQ (Kwiatkowski et al., 2019) | | | | 5-shot | sampling |
| HumanEval (Chen et al., 2021) | | | x | | sampling |
| MBPP (Austin et al., 2021) | | | x | 3-shot | sampling |

Table 6: Downstream benchmarks considered in this work. We report the average performance over *All* of these tasks, along with average performance over the *Core*, *Math*, and *Code* subsets. We use top-1 accuracy as the metric for all non-coding tasks, where the generated prefix for sampling tasks must match the reference. For coding tasks, we consider Pass@1 as the metric.

Each point in Fig. 1 corresponds to the relative quantization error with respect to a particular subset of FFW (gating, linear, or the mean of the two) against the downstream performance for one of these models.

# D    PAIRED QUANTIZATION HYPERPARAMETER DETAILS

## D.1    ALTERNATIVE CHOICES FOR THE PSEUDO-LOSS

Initially we considered other pseudo-loss choices for learning paired transformations. Using the notation described in Section 4.2, we considered (1) the mean of squared channel-wise maxima:

$$\frac{\sum_{i=1}^{m}(M_i^u)^2}{m} + \frac{\sum_{j=1}^{n}(M_j^v)^2}{n} \frac{\sum_{i=1}^{d_1}m_u(i)^2}{d_1} + \frac{\sum_{j=1}^{d_3}m_v(j)^2}{d_3}$$

and (2) the weighted mean of squared channel-wise maxima:

$$\frac{\|V'\|_F \sum_{i=1}^{d_1}m_u(i)^2}{d_1} + \frac{\|U'\|_F \sum_{j=1}^{d_3}m_v(j)^2}{d_3}$$

This latter objective is motivated by the intuition that quantization error in one weight matrix has the potential to be magnified if the norm of its paired weight is large. Thus, quantization errors should be penalized accordingly. This objective explicitly acknowledges the fact that $U'$ and $V'$ directly interact in the computation graph. We sweep over the choice of this pseudo-loss in the following subsection.

## D.2    HYPERPARAMETER TUNING

We explored several hyperparameters for learning paired VO transformations. In this section, we present sweeps over these hyperparameters, noting which ones the PQE happened to be

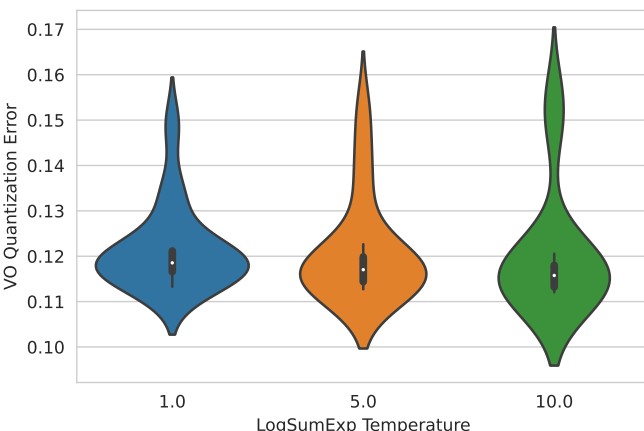

Figure 2: Distribution over the for the VO product PQE as a function of the temperature on the LogSumExp pseudo-loss, optimized with Adam. Each violin encompasses a sweep over learning rate in $10^{\{-4,-3,-2,-1\}}$, and orthogonal regularization weight in $\{0, 0.1, 1\}$ for the given value of $t$. Note that we exclude runs with learning rate of 1.0, as these runs diverged.

sensitive to. We fix number of iterations to 100,000 for all runs, and consider Cayley SGD (with $\beta = 0.1$ momentum) as well unconstrained optimization with Adam ($\beta = 0.1$), learning $R$ to transform V/O for all layers in a Gemma2-2B-PT model. We compute the mean PQE across all layers, assuming uniform channel-wise 4-bit quantization for $\widehat{\cdot}$. Note that we report the relative PQE in this section – relative to $\|W_v^T W_o\|_F$. This ensures that average PQE is not unduly influenced by the magnitude of $W_v^T W_o$ for any given layer.

We varied:

- base learning rate $lr \in 10^{\{-4,-3,-2,-1,0\}}$
- $\lambda_{orth} \in \{0, 0.01, 0.1, 1\}$
- $t \in \{1, 5, 10\}$
- Pseudo-loss $\in \{\text{LogSumExp, Sum of Squares, Weighted Sum of Squares}\}$

**LogSumExp temperature**   PQE as a function of $t$, the temperature of the LogSumExp pseudo-loss, is shown in Fig. 2. While the mean is relatively unchanged across settings of $t$, the variance of the ultimate PQE increases with $t$. We chose $t = 5$ for LogSumExp pseudo-loss experiments, as PQE was relatively insensitive to this hyperparameter.

**Learning rate & Pseudo-loss**   While $t$ had little effect on PQE, both the base learning rate and choice of pseudo-loss affected the stability of the learning curve. Fig. 3 displays learning curves for Cayley SGD and Adam as a function of learning rate and pseudo-loss.

PQE tends to converge quickly with Adam assuming that the learning rate is small enough ($< 0.1$). This is true irrespective of the choice of pseudo-loss. While Cayley SGD also smoothly reduces PQE, it benefits from a higher learning rate for PQE to converge in the allotted iterations. In general, the weighted sum of squares (SUM_SQ_WEIGHTED) objective yields the lowest PQE, with poor convergence exhibited by the LogSumExp (MAX) and sum of squares (SUM_SQ) objectives.

**Importance of learning an orthonormal** $M$   Although prior work has learned $M$ such that $M$ is a rotation, in practice, $M$ need only be invertible to ensure computational invariance. To determine how important it is that $M$ be orthonormal, we select the minimum PQE run per objective, optimization method, and orthonormal regularization weight, while sweeping over base learning rate, as well as $t$ for the LogSumExp runs.

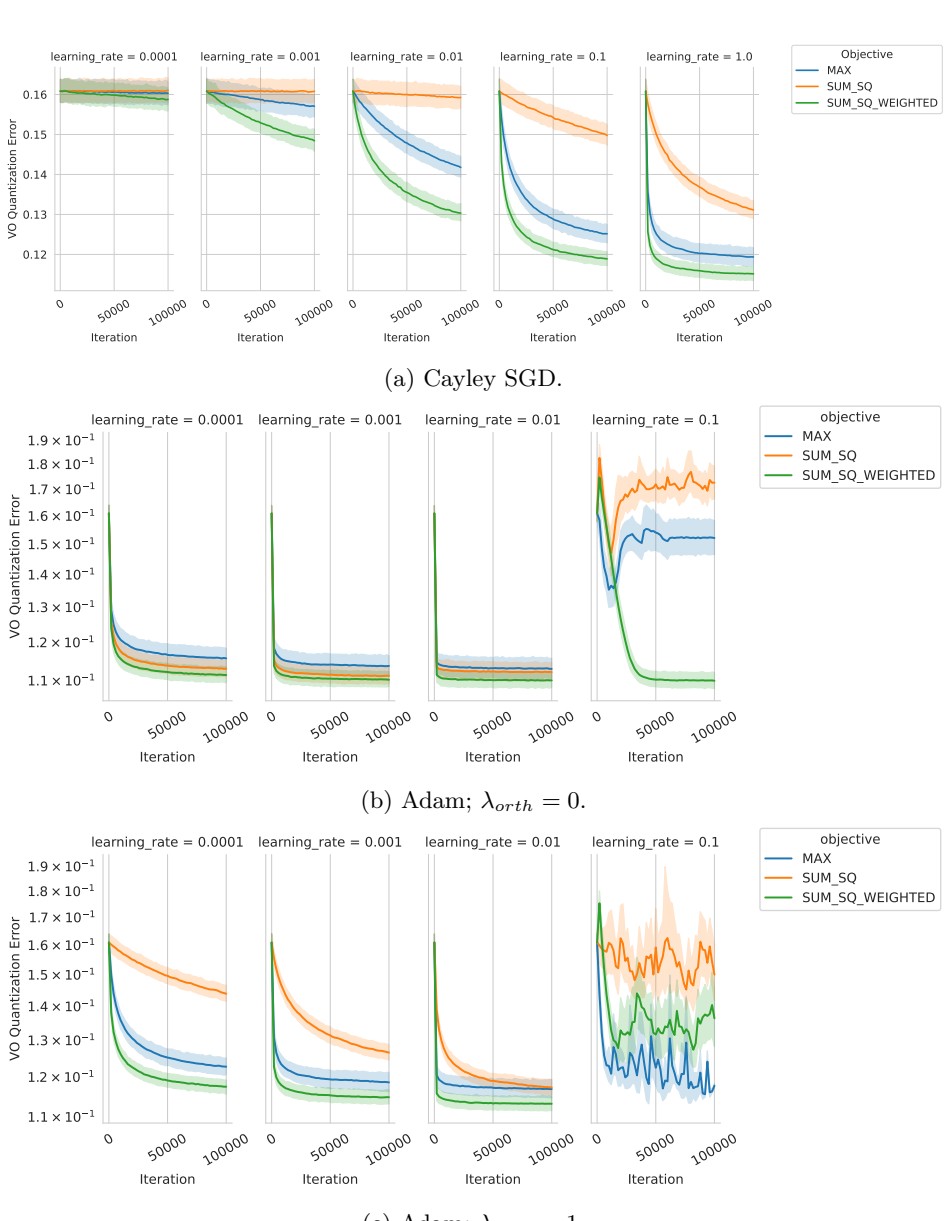

(a) Cayley SGD.

(b) Adam; $\lambda_{orth} = 0$.

(c) Adam; $\lambda_{orth} = 1$.

Figure 3: Learning curves for Cayley SGD (top) and Adam with various orthonormal regularization weights (center and bottom), with each pseudo-loss a separate line. Learning rate is varied along columns. PQE is on the y-axis and iteration count on the x-axis. Each line corresponds to the mean PQE across all model layers for a given pseudo-loss, with the 95% bootstrap confidence interval indicated by the shaded region.

| Objective | Opt | $\lambda_{\text{orth}}$ | PQE | $\min \sigma$ | $\max \sigma$ |
|---|---|---|---|---|---|
| SumSqWted | Adam | 0.00 | 0.110 | 338.848 | 894.407 |
| LogSumExp | Adam | 0.10 | 0.118 | 0.978 | 1.062 |
| LogSumExp | Adam | 0.01 | 0.118 | 0.845 | 2.175 |
| LogSumExp | Adam | 0.00 | 0.120 | 0.876 | 2.508 |
| LogSumExp | Adam | 1.00 | 0.120 | 0.981 | 1.049 |
| SumSqWted | Cayley | 0.00 | 0.134 | 0.992 | 1.000 |
| LogSumExp | Cayley | 0.00 | 0.142 | 0.997 | 1.001 |
| SumSq | Cayley | 0.00 | 0.152 | 0.999 | 1.001 |
| SumSqWted | Adam | 0.01 | 0.166 | 2.503 | 912.959 |
| SumSqWted | Adam | 0.10 | 0.185 | 1.915 | 905.954 |
| SumSqWted | Adam | 1.00 | 0.200 | 1.460 | 730.075 |
| SumSq | Adam | 0.10 | 0.255 | 0.867 | 165.328 |
| SumSq | Adam | 0.01 | 0.261 | 0.886 | 222.902 |
| SumSq | Adam | 1.00 | 0.462 | 0.869 | 138.006 |
| SumSq | Adam | 0.00 | 0.639 | 0.551 | 971.175 |

Table 7: Relative mean PQE across all layers after 100K iterations, as a function of objective, optimization method, and $\lambda_{orth}$. The average minimum/maximum eigenvalue across all layers for the learned $R$ are given as well.

Table 7 displays the relative PQE for each of these runs along with the minimum and maximum eigenvalue, averaged across all layers, for the learned $M$. The lowest PQE transformations are learned through unconstrained, Adam, optimization without any weight on the orthonormal regularization term, with a mean relative PQE of 0.110 vs. 0.118 for the next best run. Note also that the average minimum and maximum eigenvalue for these transformations are quite large, suggesting the learned transformations are far from rotations.It is important to also note that learning strict rotation transformations via Cayley SGD yields mean PQE which is at least 21.7% higher than the best unconstrained learning run.

Note that PQE is an intrinsic measure of quality, and one must also evaluate these quantized models downstream to note any degradation. In terms of stability with respect to $\lambda_{\text{orth}}$, learning transformations that are close to rotation, and achieving low PQE, unconstrained optimization with the LogSumExp pseudo-loss yields a better solution than weighted sum of squares or learning $M$ using natural gradient descent. With that in mind, we selected unconstrained optimization with LogSumExp pseudo-loss unless mentioned otherwise.