# OpenReview forum: "CafeQ: Calibration-free Quantization via Learned Transformations and Adaptive Rounding"
_ICLR.cc/2026/Conference — ICLR 2026 Conference Withdrawn Submission_

### Official Review · Reviewer_ZLs1 · 2025-10-17

**Soundness:** 2
**Presentation:** 1
**Contribution:** 1
**Rating:** 2
**Confidence:** 3

**Summary:**

This paper studies post-training quantization for LLMs with the need of calibration data. The key idea is to optimize
transformations and adaptive rounding by proxy design for the quantization loss. The reported experimental results on Gemma 2 models (2B and 9B) have shown promising performance of the proposed method.

**Strengths:**

+ Originality: The calibration-free PTQ has its merit since the majority of existing methods require some form of calibration data.
+ Quality: the reported experimental results in Tables 3-5 show improved performance over baseline methods including Uniform and Random.
+ Clarity: the majority of the paper is easy to follow.
+ Significance: PTQ for LLMs remain a timely hot topic in AI research.

**Weaknesses:**

-My major concern is about technical novelty and experimental verification. The key ideas presented in this work largely exist in the literature, and the authors did not articulate the motivation behind their approach (other than calibration-free).
-The two claims (about the "central questions in scalar PTQ for LLMs") at the end of the intro. section as well as the listed contributions in Sec. 3 often lack substantial justification. For example, the author(s) claimed "three primary contributions" on Line 135 but there are four listed contributions (bold-typed headers) in the following paragraphs. Such a lack of consistency might be tied to the next weakness.
- Poor literary presentation. The paper can be significantly revised, especially the meat of this paper (Sec. 4 and Sec. 5). Many mathematical formulas are presented without explaining the underlying motivation. Meanwhile, those equations (mostly occupying the entire row) take a lot of space, which is not a good idea due to the 9-page limit of ICLR submissions. Accordingly, the conclusion section is too short. Authors need to pay closer attention to a more efficient use of the space while revising this paper.

**Questions:**

1. How does the proposed approach compare against SmoothRot [1]?
2. Why do you only report results for Gemma 2 models? Many other researchers have chosen LLAMA 2/3 models instead.
3. You have included many related work in Appendix A. But your experimental results did not include comparison against any of them. How can you justify your work advance the SOTA?

[1] https://arxiv.org/pdf/2506.05413

---

### Official Review · Reviewer_dCYq · 2025-10-26

**Soundness:** 2
**Presentation:** 2
**Contribution:** 2
**Rating:** 4
**Confidence:** 5

**Summary:**

This paper presents CafeQ, a calibration-free post-training quantization method for Large Language Models.
The core idea is to address the dual challenges of outlier handling and rounding in the absence of calibration data by learning input-weight transformations.
The key contributions are the use of a Frobenius norm proxy loss, learned block-diagonal transformations for single matrices, and a novel combination of computational invariance and adaptive rounding for coupled matrix pairs.
The method is evaluated on Gemma 2 models, showing consistent improvements over calibration-free baselines and competitive performance with the calibration-based GPTQ, all while adding minimal inference overhead.

**Strengths:**

The paper clearly articulates the critical need for calibration-free quantization in real-world scenarios (privacy, data scarcity, domain shift).

The improvements over uniform and random rotation baselines on Gemma2 are clear and consistent, especially for the challenging 3-bit case.

**Weaknesses:**

1) The work demonstrates limited novelty and insufficient distinction from prior work. The core idea of learning a transformation (M) to improve quantization is from the central contribution of SpinQuant (Liu et al., 2024). The distinction here is only the use of a proxy loss instead of calibration data and the relaxation of orthonormality constraints. The use of (W1 M^{-1})(M W2) is explicitly discussed and utilized in related works such as QuaRot (Ashkboos et al., 2024b).
As for adaptive rounding, the proposed alternating rounding is interesting. However, the idea of joint rounding to minimize the error of a matrix product is a logical extension of GPTQ-style Hessian-based rounding. The novel aspect is doing this without a Hessian (using the pseudo-inverse), but the connection and distinction should be more clearly discussed.

2) All experiments are conducted exclusively on the Gemma 2 family. The LLM ecosystem is diverse (e.g., LLaMA, Qwen), with different architectures, training data, and outlier characteristics. The claim of a general method requires validation across at least one more architecturally distinct model family.

3) The entire framework relies on the L2 norm of the weight reconstruction error being a good proxy for task performance. While Figure 1 shows a correlation, it is not perfect (Spearman ~ -0.64).

4) The paper rightly highlights low inference overhead. However, the offline cost of the optimization is substantial. Learning transformations for every layer via gradient descent (for single matrices) and running iterative SVD-based rounding (for coupled matrices) is computationally expensive. For a one-time quantization, this may be acceptable, but it should be explicitly discussed and quantified, as it affects the practicality of the method for very large models or frequent re-quantization.

**Questions:**

1) The proposed method performs well on Gemma 2, but does it work equally well on other architectures like GLM, Qwen, or MoE models?

2) You applied paired quantization only to (W_v, W_o). What was the performance when applied to (W_q, W_k) in layers without RoPE?

3) Can you provide an analysis or discussion of cases where minimizing the Frobenius proxy loss fails to improve downstream accuracy?

---

### Official Review · Reviewer_7ejg · 2025-11-01

**Soundness:** 3
**Presentation:** 3
**Contribution:** 3
**Rating:** 6
**Confidence:** 4

**Summary:**

This paper introduces **CafeQ**, a calibration-free post-training quantization (PTQ) method for large language models (LLMs). The approach aims to mitigate the performance degradation typically caused by weight outliers and suboptimal rounding in uniform quantization, without relying on calibration data. This is a well-written and technically sound paper that makes a significant contribution to the field of efficient LLM inference. It tackles the important and challenging problem of calibration-free quantization with a novel and comprehensive set of methods. The empirical results are compelling, showing clear improvements over existing data-free baselines and even challenging a leading calibration-based method. The weaknesses are relatively minor and point to interesting future directions rather than fundamental flaws.

**Strengths:**

S1.  Addresses a significant practical limitation of many PTQ methods by eliminating the need for calibration data, enhancing applicability in data-scarce or privacy-sensitive scenarios.
S2.  The use of the Frobenius norm of the reconstruction error as a proxy loss is well-motivated and empirically validated (via Spearman correlation), providing a principled foundation for the calibration-free optimization.
S3.  The paper presents a cohesive framework addressing both outlier mitigation (via learned transformations) and rounding, for both isolated and coupled weight matrices.
S4. The design carefully considers inference overhead. Structured (block-diagonal) transformations keep computational costs low, and the paired matrix technique introduces zero additional inference cost.
S5.  Demonstrates significant and consistent improvements over strong calibration-free baselines (e.g., raising the 3-bit Gemma 2 9B score from 52.0 to 60.6). The ablation studies are thorough and validate the contribution of each component.
S6.  Achieving performance comparable to GPTQ is a notable result, highlighting the potential of learned, data-free methods.

**Weaknesses:**

**W1**.  The paired quantization technique is primarily applied to the $W_v, W_o$ pair due to incompatibility with Rotary Positional Embeddings (RoPE) in the $W_q, W_k$ pair. This limits its applicability in many current architectures that still rely on RoPE, though the authors correctly note the trend toward RoPE-free models. Of course, I don't think this issue is significant.
**W2**.  The learning of transformation matrices and the adaptive rounding algorithm incur non-trivial offline computation costs. The paper does not provide a detailed analysis of this optimization time.
**W3**.  While the correlation between the proxy loss and downstream performance is shown for FF layers, a broader validation across all layer types would strengthen the core assumption.
**W4**. The appendix reveals that the optimization of paired matrices can be sensitive to hyperparameters like the learning rate and the choice of pseudo-loss. This might affect the method's ease of adoption.
**W5**. When this paper is accepted, it should be cited using `\citep` instead of `\cite` to cite articles; this is stated on the official website.
**W6**. Matrix should be bolded using `\mathbf`.

**Questions:**

**Q1**.  Could the authors provide more details on the total wall-clock time and computational resources required to learn the transformations for a model like Gemma 2 9B? How does this cost scale with model size?
**Q2**.  The authors demonstrated a strong correlation between the $\ell_2$ proxy loss and downstream performance for FF layers. Did you observe a similarly strong correlation for attention layer weights $W_q, W_k, W_v, W_o$? If not, how does this impact the overall framework's robustness?
**Q3**.  For architectures that do use RoPE, have you explored any approximations or alternative strategies to apply your powerful paired quantization technique to the $W_q, W_k$ matrices, perhaps by modifying the RoPE computation?
**Q4**.  The authors position your method as orthogonal to GPTQ. Have you conducted any preliminary experiments on combining CafeQ's learned transformations with GPTQ's Hessian-based rounding? Could this hybrid approach push the state-of-the-art for calibration-*allowed* scenarios?

---

> ### Comment · Reviewer_7ejg · 2025-11-19
> **Tips**
>
> Dear authors,
> if you don't have time to update the paper, just put the rebuttal in the comment. I acknowledge the workload is heavy. I will change the score according to your response. Your paper is very good.

---

> ### Comment · Reviewer_7ejg · 2025-11-25
> **I need to reply**
>
> The PC reminds me to reply, and I also want to reply. However, the author didn't reply to me, and I don't know how to reply.

---

### Official Review · Reviewer_gfUr · 2025-11-03

**Soundness:** 2
**Presentation:** 3
**Contribution:** 2
**Rating:** 2
**Confidence:** 5

**Summary:**

The paper proposes CafeQ, a post-training quantization method that uses uniform adaptive rounding after a intelligently-selected block diagonal linear transformation.

**Strengths:**

Quantization is a very important subject, and the learned linear transformation idea is interesting! The tech in 4.2 and 5.1 does seem to have some promise.

**Weaknesses:**

The standard method for quantizing weight matrices in LLMs is not uniform quantization. It's floating-point quantization. Float8, MXFP4, NVFP4, etc are all floating point quantization formats. "The rationale for choosing uniform quantization is because it is highly efficient and broadly supported by most modern hardware accelerators" -- most modern hardware accelerators support floating-point quantization.

>Calibration data-free...representative data for calibration is unavailable... even when data exists, its use may be prohibited due to privacy and security concerns...relying on a static calibration set introduces a significant vulnerability to domain shift...may be prohibitive.

This whole paragraph seems like an overclaim. The first citation does not assert that in low-resource languages, a calibration set is not feasible. The second citation is to ZeroQ, which says "Quantization is a promising approach for reducing the inference time and memory footprint of neural networks. However, most existing quantization methods require access to the original training dataset for retraining during quantization. This is often not possible for applications with sensitive or proprietary data," But what ZeroQ is saying is problematic is accessing the original training set; this is not what is required for a devset-based quantization method like GPTQ, where just _some data_ is needed. And the third claim about domain shift is an empirical claim that seems to just be unsupported.

You gotta compare to more recent quantization methods than GPTQ! A bunch of them are cited in the intro: compare to more of those.

Please break down the accuracy results in Table 3 by task.

**Questions:**

How would this approach compare to a modern PTQ method like QTIP or AWQ or SpinQuant?

---

### Comment · Area_Chair_nJEF · 2025-11-28

Dear authors and reviewers,

Please remain professional and refrain from being influenced by the event. If anyone violates the rules, please let me know, and I will flag and report it to the Program Chairs.

Your AC

---

> ### Comment · Reviewer_7ejg · 2025-11-28
> **Rebuttal Require**
>
> I really want to see the author's rebuttal, but I haven't seen it until today. If you can remind the author to reply to the rebuttal.
>
>
> REVIEWER 7ejg

---

> > ### Comment · Area_Chair_nJEF · 2025-11-28
> >
> > The rebuttal is optional, not required.

---

### Note · Authors · 2026-01-01

**Comment:**

Dear Area Chair and Reviewers,

We have decided to withdraw our paper.

We sincerely thank the reviewers for their detailed comments and acknowledging the strengths of the paper. We believe the paper could benefit from incorporating the feedback provided—particularly regarding experimental settings and baseline comparisons. We intend to incorporate these suggestions to strengthen the paper for a future submission.

Best regards,
The Authors

**Withdrawal Confirmation:**

I have read and agree with the venue's withdrawal policy on behalf of myself and my co-authors.